# Vitamin D Levels as an Important Predictor for Type 2 Diabetes Mellitus and Weight Regain Post-Sleeve Gastrectomy

**DOI:** 10.3390/nu14102052

**Published:** 2022-05-13

**Authors:** Alanoud Aladel, Alice M. Murphy, Jenny Abraham, Neha Shah, Thomas M. Barber, Graham Ball, Vinod Menon, Milan K. Piya, Philip G. McTernan

**Affiliations:** 1Community Health Sciences, College of Applied Medical Sciences, King Saud University, Riyadh 11451, Saudi Arabia; aaladel@ksu.edu.sa; 2Department of Biosciences, School of Science and Technology, Nottingham Trent University, Nottingham NG1 8NS, UK; alice.murphy@ntu.ac.uk (A.M.M.); graham.ball@ntu.ac.uk (G.B.); 3Warwickshire Institute for the Study of Diabetes Endocrinology and Metabolism, University Hospitals Coventry and Warwickshire, Clifford Bridge Road, Coventry CV2 2DX, UK; jenny.abraham@uhcw.nhs.uk (J.A.); t.barber@warwick.ac.uk (T.M.B.); vinod.menon@uhcw.nhs.uk (V.M.); 4Jeffrey Kelson Centre, Central Middlesex Hospital, Acton Lane, London NW10 7NS, UK; neha.parekh@nhs.uk; 5Division of Biomedical Sciences, Warwick Medical School, University of Warwick, Gibbet Hill, Coventry CV4 7AL, UK; 6Division of Health Sciences, Warwick Medical School, University of Warwick, Gibbet Hill, Coventry CV4 7AL, UK; 7School of Medicine, Western Sydney University, Campbelltown, NSW 2560, Australia; m.piya@westernsydney.edu.au; 8South Western Sydney Metabolic Rehabilitation and Bariatric Program (SWS MRBP), Camden and Campbelltown Hospitals, Camden, NSW 2570, Australia

**Keywords:** diabetes, type 2 diabetes, diabetes remission, bariatric, sleeve gastrectomy, vitamin D, obesity, weight regain

## Abstract

Weight Loss Surgery (WLS), including sleeve-gastrectomy (SG), results in significant weight loss and improved metabolic health in severe obesity (BMI ≥ 35 kg/m^2^). Previous studies suggest post-operative health benefits are impacted by nutrient deficiencies, such as Vitamin D (25(OH)D) deficiency, while it is currently unknown whether nutrient levels may actually predict post-surgery outcomes. As such, this study investigated whether 25(OH)D levels could predict metabolic improvements in patients who underwent SG. Patients with severe obesity (*n* = 309; 75% female) undergoing SG participated in this ethics-approved, non-randomized retrospective cohort study. Anthropometry, clinical data, 25(OH)D levels and serum markers were collected at baseline, 6-, 12- and 18-months post-surgery. SG surgery resulted in significant improvements in metabolic health at 6- and 12-months post-surgery compared with baseline, as expected. Patients with higher baseline 25(OH)D had significantly lower HbA1c levels post-surgery (*p* < 0.01) and better post-surgical T2DM outcomes, including reduced weight regain (*p* < 0.05). Further analysis revealed that baseline 25(OH)D could predict HbA1c levels, weight regain and T2DM remission one-year post-surgery, accounting for 7.5% of HbA1c divergence (*p* < 0.01). These data highlight that higher circulating 25(OH)D levels are associated with significant metabolic health improvements post-surgery, notably, that such baseline levels are able to predict those who attain T2DM remission. This highlights the importance of 25(OH)D as a predictive biomarker of post-surgery benefits.

## 1. Introduction

Obesity is a serious public health challenge affecting more than 650 million adults worldwide [1] noting that class 3 obesity (BMI > 40 kg/m^2^) is responsible for a reduction in life expectancy of at least 10 years [2]. Obesity is also the principal risk factor in the development of type 2 diabetes mellitus (T2DM) and contributes to cardiovascular disease (CVD) progression and certain cancers [3,4]. Whilst diet, physical exercise, medication and behavioral therapy can support alleviating early weight gain, the development of severe and class 3 obesity (BMI ≥ 35 kg/m^2^ and BMI > 40 kg/m^2^, respectively) presents additional challenges that are difficult to overcome through these measures. As such, WLS is considered an effective treatment option for cases of severe obesity, as it achieves a relatively rapid weight reduction that is often maintained long-term, contributing to the remission of several comorbidities [5,6,7] and subsequently reducing the mortality rate [8].

As determinants to improve the success rate of WLS, BMI and smoking status are monitored, with pre-surgical weight loss and smoking cessation often required before surgery will be undertaken. These required interventions have been linked to favorable outcomes post-surgery, including increased weight loss and reduced risk of ulceration and infection post-surgery [9,10]. With these interventions in place, bariatric surgical procedures achieve a high success rate, with 50–60% of patients achieving and maintaining weight loss 5-years post-surgery. However, in the remaining 40% of patients, there are a number who experience weight regain following an initial weight loss, returning to their original body mass index (BMI) within 5-years post-surgery [11].

The percentages of patients with T2DM who enter remission post-bariatric surgery vary between studies, ranging from around 40 to 85% [12,13,14,15]. One of the strongest predictors of post-surgery remission is the duration of T2DM [16]; however, this factor cannot easily be measured by clinicians or patients due to delays in diagnosis being common. As such, studying potential biomarkers which may predict T2DM remission is necessary.

It is understood that nutrient deficiencies are a common complication of WLS, particularly with the malabsorptive procedures Roux-en-Y gastric bypass (RYGB) and mini gastric bypass-one anastomosis gastric bypass (MGB-OAGB), and several studies have demonstrated the prevalence of nutritional deficiencies among patients with obesity prior to WLS [17,18,19,20,21,22]. Insufficient (25–50 nmol/L) or deficient (<25 nmol/L) levels of circulating Vitamin D (25(OH)D) is one of the most common nutrient deficiencies with obesity, and this is commonly associated with T2DM status [17,23]. Several studies have demonstrated the positive role of 25(OH)D in adipose tissue metabolism, such as influencing pre-adipocyte differentiation [24,25] or enhancing insulin signaling pathways and insulin secretion [26]. For instance, Vdr knockout mice displayed a reduction in fat tissue mass, lower circulating leptin levels, and higher food intake [27]. Similar effects were observed in mice that were unable to produce 1,25(OH)2D due to a lack of Cyp27b1 expression and had similar results [28]. These findings strongly suggest that vitamin D has a role in lipid accumulation in adipose tissue. Moreover, vitamin D could restore impaired insulin signaling through the inhibition of signaling pathways (including NF-κB, SCAP/SREBP and CML/RAGE cascades) [29]. Vitamin D may also regulate intracellular Ca^2+^ concentration and participates in insulin secretion by pancreatic β-cells [30]. As such, subjects with increasing weight gain and aligned 25(OH)D deficiency have an increased risk of developing T2DM [31,32,33]. Currently, 25(OH)D deficiency is observed in 94% of people with obesity who are candidates for WLS [34] and it is very common post-WLS as a result of multiple factors including the type of the surgery, vitamin D intake, sunlight exposure, and season [35]. This deficiency has been shown to influence post-operative outcomes including T2DM, dyslipidemia, high blood pressure, metabolic syndrome and weight [36,37]. Previous interventional studies have highlighted that 25(OH)D supplementation is associated with weight loss and improved β cell function amongst subjects with obesity and early T2DM subjects [37,38]. However, the use of physiological 25(OH)D levels as a predictor of positive metabolic outcomes, such as T2DM remission and weight loss following sleeve gastrectomy (SG) is currently unknown. As such, this study examined whether 25(OH)D levels in patients undergoing bariatric surgery may predict metabolic improvements post-SG.

## 2. Materials and Methods

### 2.1. Subjects and Study Design

A non-randomized, retrospective cohort study was conducted on patients with severe obesity (*n* = 309; 75% female) undergoing SG at Warwickshire Institute for the Study of Diabetes, Endocrinology and Metabolism (WISDEM), between 2010 and 2018. A multidisciplinary weight management team evaluated all patients before and after SG. All patients were evaluated pre-operatively and post-operatively at 3, 6, 12 and 18 months. The dropout rate at 18 months was 36% (113/309). During the pre-operative period, nutrition advice was given to patients to modify their eating patterns, encouraging a minimum goal of 5% excess body weight loss prior to surgery (details in Appendix A).

### 2.2. Anthropometrics, Clinical Data and Blood Biochemistry

Appointments by the multidisciplinary clinical team included anthropometric measurements, laboratory tests, blood pressure measurements, and clinical assessment of any complications or comorbidities to adjust the intervention and to detect any improvement or remission. The percentage of excess weight loss (%EWL) was calculated by defining excess weight as anything over that which corresponded to a BMI of 25 kg/m^2^ for each subject. Blood samples were collected and measured at baseline, 6- and 12-months post SG (details in Appendix A). Anti-diabetic medication was recorded before and after SG, and patients with different comorbidities were identified by a formal letter from the physician and/or endocrinologist. T2DM remission was defined using a modification of the American Diabetes Association Consensus Group as HbA1c < 6.5% (48 mmol/mol) (or fasting glucose < 6.9 mmol/L if HbA1c was not available), as well as the absence of any active pharmacological therapy for diabetes [39]. Baseline demographics for patients with and without T2DM are shown in Appendix A.

All blood biochemistry measurement techniques are provided in detail in the Appendix A.

### 2.3. ANN Model Development and Sensitivity Analysis

Artificial Neural Network (ANN) models were developed in NeuroSolutions (NeuroDimension Ltd., Gainesville, FL, USA) using Monte Carlo Cross validation, early stopping on a cross-validation data set, a Levenburg-Marquardt algorithm and a TanH activation function. The number of hidden units was optimized based on test data. (Further detail in Appendix A). Sensitivity analysis was carried out on the Trained Optimized Model by applying variables to a trained model to determine the factors that contribute to that model, as well as how much impact they have on the model.

### 2.4. Statistical Analysis

Statistical analyses were performed using SPSS 25.0 (IBM, Armonk, NY, USA). Categorical data are presented as percentages and continuous variables are reported as mean ± standard deviation unless the standard error of the mean (SEM) is stated. Data were examined for normality according to the Shapiro–Wilks criteria and by visual inspection of QQ-plots. Non-normally distributed variables were logarithmically transformed (natural logarithm) before use in parametric analyses. Comparisons between categorical groups and percentages were calculated by Chi-Square (χ^2^) and McNemar tests. For continuous variables, one-way ANOVA was used to assess differences between multiple groups. Two-tailed independent samples *t*-tests or Wilcoxon Signed-Rank Tests were used to compare mean differences between two independent groups for parametric and non-parametric data, respectively. Mean differences between pre- and post-values of the same variables were determined using a 2-tailed paired *t*-test. Bivariate Pearson correlation analysis was used to analyze correlations between parametric variables. Simple and multiple linear regression analysis was used to assess the effects of single or multiple factors on post-operative independent variables. A multiple binary logistic regression analysis (forward method) was performed with the incidence of post-operative T2DM as the binary outcome variable and 25(OH)D, FPG and age as the explanatory variables.

## 3. Results

### 3.1. Baseline Demographics for Patients

More than 47% of patients were diagnosed with T2DM at least one year prior to surgery. No significant differences were observed between the groups in any of the baseline anthropometric data (Table 1). Hypertension, T2DM and gastroesophageal reflux disease (GORD) were the top three most dominant comorbidities with significant remissions post-operatively (Appendix A). ANOVA revealed no overall significant differences in seasonal variation between circulating 25(OH)D collected at different times of the year (*p* = 0.111).

### 3.2. Metabolic Health Improves Post-SG

Indicators of metabolic syndrome were assessed pre- and post-surgery. Post-surgery, over 51% (75/146) of T2DM patients were in remission, defined as HbA1c < 6.5% (48 mmol/mol) and an absence of active pharmacological therapy for diabetes. T2DM and non-T2DM participants exhibited significant improvements in several clinical and biological indicators of metabolic health at 6- or 12-months post-surgery compared to baseline regardless of their diabetes status (*p* < 0.05). As expected, many components of metabolic syndrome were statistically different between T2DM and non-T2DM subjects at baseline, 6- and 12-months post-surgery (*p* < 0.05; Table 2). Fasting plasma glucose and HbA1c levels were significantly decreased post-surgery for both T2DM and non-T2DM subjects (*p* < 0.01; Figure 1A,B).

The reduction in fasting glucose and HbA1c was reflected in a significant reduction in anti-diabetic medications one-year post-surgery (*p* < 0.001; Figure 2A). The majority of participants with T2DM (123/146, 84.2%) were prescribed metformin, with a dose ranging from 500–3000 mg/day, whilst the few remaining patients were on other antidiabetic medications such as insulin and sulphonylureas. Prior to the surgery, 12% of patients with T2DM were on insulin medication (18/146); post-surgery 33% of these (6/18) were in remission and were no longer taking insulin; of those not in remission 50% lowered their dosage or switched to metformin (6/12). Patients who used insulin and were in remission post-surgery had significantly higher levels of 25(OH)D compared with patients who used insulin but were not in remission (48 ± 18 vs. 18.6 ± 6 nmol/L, *p* < 0.05).

In order to analyze changes in medication dosage, patients with T2DM were divided into two categories—those who increased or did not change the dose and those who reduced the dose or stopped the medications. One-year post-surgery, 81% of participants had decreased the dose or stopped the medication (*p* < 0.001; Figure 2B). As such, these data demonstrate that the cohort in this study reflects the general population in regards to the biological differences observed between T2DM and non-T2DM subjects, as well as improvements following WLS.

### 3.3. Impact of 25(OH)D Supplementation Varies Depending on T2DM Status

As insufficient (25–50 nmol/L) or deficient (<25 nmol/L) circulating 25(OH)D levels were prevalent in the majority of participants (68.6%) at baseline, all participants were prescribed 25(OH)D supplementation post-surgery as part of routine care, which resulted in a marked decrease in cases of insufficiency and deficiency at 6-months and one-year post-surgery (15% and 16.7%, respectively). Interestingly, although both groups were prescribed 25(OH)D supplementation, a higher percentage of 25(OH)D insufficiency/deficiency was observed in T2DM participants one-year post-surgery (20.5%) compared with those without T2DM (12.9%) (*p* < 0.05; Figure 3), despite a similar percentage at baseline and no significant differences in BMI and EWL between the groups.

### 3.4. 25(OH)D Levels Negatively Correlate with Post-Surgery BMI, Weight and Weight Regain

Pre-surgery 25(OH)D levels were assessed to determine if they correlated with BMI, weight and excess weight loss (EWL). Pre-surgery 25(OH)D levels were negatively associated with pre- and up to 6-months post-surgical BMI (*p* < 0.05). Similarly, pre-surgery 25(OH)D levels were negatively associated with pre- and up until 9-months post-surgical weight (*p* < 0.05). No significant correlations were observed between baseline circulating 25(OH)D and EWL at any time points.

Interestingly, 25(OH)D levels at 6 months were positively correlated with EWL at 6- and 9-months post SG (r = 0.187, r = 0.170, respectively; *p* < 0.05). Levels of 25(OH)D at 12 months were also positively correlated with EWL at 12- and 18-months post-SG (r = 0.149, r = 0.154, respectively; *p* < 0.05).

In addition to these findings, circulating 25(OH)D levels at 12 months were negatively correlated with weight regain 18-months post-surgery (*p* < 0.05; r = −198). Along with fasting glucose levels at 12 months, multiple regression analysis (using the stepwise method) has shown that circulating 25(OH)D levels at 12 months were responsible for 10.4% of the divergence in weight regain at 18 months and this percentage was even higher (14.6%) among patients with T2DM (Appendix A). These data suggest levels of circulating 25(OH)D at 12 months may be involved in the prevention of weight regain at 18 months.

Sensitivity analysis revealed that circulating 25(OH)D levels at baseline, 6 months and 12 months were all within the top 12 predictors of weight regain out of 46 variables tested. Circulating 25(OH)D at 6-months post-surgery was the fifth-best predictor with a sensitivity of 1.5, baseline 25(OH)D was the eighth-best predictor with a sensitivity of 1.2, and circulating 25(OH)D at 12-months post-surgery was the twelfth-best predictor with a sensitivity of 1.0 (Appendix A).

### 3.5. Baseline 25(OH)D Predicts HbA1c Levels at Baseline and One Year Post-Surgery

The ability to predict HbA1c levels at baseline and one-year post-surgery was assessed via bivariate correlation and linear regression analysis. A significant negative correlation was observed between baseline 25(OH)D and HbA1c levels at both baseline (*p* < 0.05) and 12 months (*p* < 0.01) post-surgery (Figure 4A,B). However, separating the cohort based on T2DM status indicated that this significance was only observed in T2DM participants (*p* < 0.05; Figure 4C,D). Moreover, no significant relationships were determined between levels of 25(OH)D at 6 or 12 months with HbA1c levels at 12 months, regardless of analyzing the cohort as a whole or separated based on T2DM status.

In addition to this, baseline 25(OH)D was shown to account for 3% of HbA1c divergence one-year post-surgery (*p* < 0.01). After separating the cohort based on T2DM status the regression was stronger for participants with T2DM, showing that 25(OH)D was responsible for 7.5% of HbA1c divergence (*p* < 0.01) whilst the regression for the non-T2DM cohort lost its significance.

### 3.6. Baseline 25(OH)D Predicts T2DM Medication Intake One Year Post-Surgery

In order to assess whether pre-surgery circulating 25(OH)D levels can predict T2DM remission post-surgery, a chi-square test of independence was performed to examine the relationship between diabetes medication intake one-year post-surgery and 25(OH)D deficiency status. The relation between these variables was significant, (X2 (2, *n* = 94) = 8.6, *p* = 0.013), with a higher number of those with sufficient baseline 25(OH)D decreasing or stopping their medication (*n* = 26) than was expected (*n* = 21) (Figure 5A). In addition to this, a univariate logistic regression was performed to compute the probability of ceasing the T2DM medication which indicated that for every unit increase in baseline 25(OH)D levels, the participant is 4% less likely to increase or remain on the same T2DM medication dosage post-surgery (R^2^ = 0.140; OR 0.959 (0.929–0.990); *p* < 0.05). This is reflected in the higher baseline 25(OH)D levels of those who decreased or stopped their medication compared to those that maintained or increased their medication (*p* < 0.01, Figure 5B,C), suggesting that participants with T2DM are more likely to reduce or stop their medication if they had sufficient 25(OH)D levels at baseline.

### 3.7. Baseline 25(OH)D Is the Second Most Relevant Factor for T2DM Remission Post-Surgery

Differences between pre-surgery factors in participants who did or did not experience diabetes remission at 12 months were investigated to initially identify which factors may impact remission. 25(OH)D, fasting plasma glucose (FPG), HbA1c, aspartate transaminase (AST), metformin dosage and presence of hypertension were all significantly different between the groups (Table 3). When considered individually using univariate logistic regression, these factors were all noted to influence T2DM remission (Table 4). The most relevant factor was the pre-surgery levels of HbA1c (R^2^ = 0.330; OR 1.079 (1.041–1.098); *p* < 0.001), with the results indicating that lower pre-surgery HbA1c levels significantly favored T2DM remission. 25(OH)D was noted to be the second most relevant pre-surgery factor (R^2^ = 0.278; OR 0.954 (0.920–0.971); *p* < 0.001). The effects of FPG, AST, metformin dosage and the presence of hypertension also influenced remission (Table 4). However, multivariate logistic regression identified that 25(OH)D and FPG were the only variables that predicted diabetes remission post-surgery (Table 4). HbA1c and medication intake were excluded due to their co-linearity with 25(OH)D and FPG. Interestingly, age did not predict remission, but when added to the model increased the R2 to 0.445 and the percentage of cases was correctly classified from 76% (model I) to 78% (model II) (Table 4). Despite BMI showing a significant contribution to the model’s R2 value, it did not improve the predictive value of any of the models.

Sensitivity analysis was then carried out to identify predictors of T2DM status at 12-months post-surgery. This identified baseline 25(OH)D and 25(OH)D at 6-months post-surgery as the second and third most sensitive predictors of T2DM status respectfully, following CRP which was the most sensitive predictor (Appendix A).

### 3.8. Baseline 25(OH)D as an Indirect Factor for Improvements in Glucose Levels

Baseline circulating 25(OH)D levels were examined against clinical indicators which are known to be associated with glucose levels at baseline, 6-months and 12-months post-surgery. Baseline 25(OH)D levels were negatively correlated with triglycerides (TG) at baseline (r = −0.142, *p* < 0.05) and positively correlated with high density lipoprotein (HDL) levels at baseline (r = 0.154, *p* < 0.05) and 6-months post-surgery (r = 0.162, *p* < 0.05). Moreover, baseline 25(OH)D predicted levels of baseline TG and HDL and explained 4% and 5% of divergence in circulating levels, respectively (*p* < 0.05). Finally, there was a significant correlation between baseline TG and HDL levels with glucose levels at baseline (TG: r = −0.260, *p* < 0.001; HDL: r = 0.343, *p* < 0.001), 6-months (TG: r = −0.149, *p* < 0.05; HDL: r = 0.275, *p* < 0.001) and 12-months (TG: r = −0.178, *p* = 0.01; HDL: r = 0.218, *p* < 0.01) post-surgery. Higher baseline TG levels were associated with higher glucose levels at all time points, whereas higher HDL levels pre-surgery were associated with lower levels of glucose at all time points.

## 4. Discussion

This study explored whether physiological 25(OH)D levels could be used as a predictor of metabolic improvement following SG. To address this question participants undergoing SG surgery were monitored over 18 months to assess biochemical biomarkers including circulating 25(OH)D levels, as well as post-surgical weight measurements and diabetes remission status. Data analysis from this study revealed three key findings: (1) baseline circulating 25(OH)D appears to be an important predictor for T2DM remission post-surgery; (2) 25(OH)D may be associated with reduced weight regain, and is associated with weight loss post-operatively; and (3) 25(OH)D and FPG are useful predictors for T2DM remission post-surgery.

Investigation into the potential of baseline circulating 25(OH)D as a predictor for T2DM status post-SG revealed that participants with higher baseline 25(OH)D had significantly lower HbA1c levels post-surgery and were more likely to reduce or stop their diabetic medication. This correlation was stronger in participants with T2DM, whose baseline 25(OH)D accounted for 8% of divergence in one-year post-surgery HbA1c levels, compared with 3% among the total cohort population. Correlations between baseline 25(OH)D and HbA1c, HOMA-IR and anti-diabetic medication intake have been shown previously; however, these studies did not investigate the impact of WLS [31,32,33,39]. These data suggest an important role for 25(OH)D in the regulation of T2DM.

Baseline circulating 25(OH)D levels were negatively associated with weight and BMI up to 9-months post-surgery; however, no correlations were observed between 25(OH)D and EWL at any time point. The association between 25(OH)D and weight regain, but not EWL, could be explained by the inhibition of adipogenesis by 25(OH)D. Studies have shown that 25(OH)D inhibits adipogenesis by suppressing adipogenic-specific genes through inhibition of PPARγ expression [40]. Further to this, recent dietary studies inducing weight loss with a very low-calorie ketogenic diet in participants with obesity also led to a significant rise in circulating 25(OH)D levels, with BMI being a strong predictor of 25(OH)D levels. Taken together, it appears that a reduction in fat mass coupled with raised (25(OH)D levels may help to further propagate the inhibition of adipogenesis to limit weight regain [41].

Interestingly, in our current study, there was also an association between increased circulating 25(OH)D at 12-months post-surgery, which resulted from the supplementation received by all participants post-surgery, and reduced levels of weight regain at the end of the study. Linear regression identified that 25(OH)D levels at 12-months post-surgery accounted for 10% of divergence in weight regained at 18-months post-surgery among all participants. Similarly, sensitivity analysis identified that circulating 25(OH)D levels at baseline, 6 months and 12 months were all within the top 12 predictors of weight regain out of 46 variables tested. These data support a longitudinal study that demonstrated 25(OH)D deficiency among patients who exhibited weight regain five-years post-RYGB, confirming that similar results are seen across these two different surgery types [11]. A possible mechanism of action for the impact of 25(OH)D on weight regain is the potential involvement of 25(OH)D in the suppression of genes involved in preadipocyte differentiation [24,42,43]. Higher 25(OH)D stores could therefore reduce the rate of adipogenesis and hence prevent weight gain.

The observed correlations between baseline 25(OH)D and (1) HbA1c levels, (2) anti-diabetic medication intakes and (3) lipid profile, may support and explain the finding that baseline 25(OH)D is able to predict the presence of T2DM one-year post-surgery. The univariate analysis determined that 25(OH)D is the second strongest pre-surgery factor that influences the rate of T2DM remission post-surgery. This provides further detail than previous work noting an association between 25(OH)D levels and the metabolic syndrome, including the presence of T2DM following gastric bypass [44,45]. Similar associations were evident in our study, where low 25(OH)D levels at baseline were apparent in 92.3% (36/39) of patients who continued to have diabetes post-surgery. Despite univariate analysis determining 25(OH)D, FPG, AST, metformin dosage and the presence of pre-surgery hypertension as useful predictors of T2DM remission, multivariate analysis highlighted only 25(OH)D and FPG as independent predictors. These factors are known to be important in influencing diabetes status [16,44]. Interestingly, age improved the multivariate analysis models despite not showing any significance to predict the remission when considered individually. However, as age is not modifiable and T2DM may occur in older ages, it may be more practical to use a model that does not involve age (model I). Alongside this, sensitivity analysis revealed 25(OH)D at baseline and 6-months post-surgery as the second and third most sensitive predictors of T2DM status 12-months post-surgery. These data constitute a good argument for monitoring 25(OH)D and glucose levels in T2DM patients prior to WLS. With regards to BMI, the cohort in this study showed no significant influence of BMI on T2DM remission, which is in agreement with previously published studies [46,47,48]; however, due to the nature of this study, only participants with severe obesity (≥35 kg/m^2^) were evaluated.

This study also highlighted the indirect modification of glucose levels by 25(OH)D at all time points, via the inverse correlation of 25(OH)D with TG, and positive correlation with HDL. Baseline levels of TG and HDL were not only strongly associated with glucose levels at all time points but were also able to be predicted by baseline 25(OH)D. Impaired lipid profiles including TG and HDL have been reported amongst patients with reduced glucose tolerance [49] and their status was found to be associated with 25(OH)D status [50]. Therefore, these results suggest that 25(OH)D plays a role in glucose homeostasis via lipid profile, providing a possible mechanism of action for the relationship between 25(OH)D and increased T2DM remission. These findings support previous work which has observed a strong negative association between 25(OH)D and TG as well as HbA1c. The prior study identified an association between 25(OH)D and HbA1c was most pronounced in individuals with TG above the upper reference limit [51]. Additionally, impaired levels of HDL and TG were able to be modified by 25(OH)D supplementation [52,53], further highlighting the indirect role of 25(OH)D in glucose homeostasis through modulating the lipid profile [54,55].

This study had certain limitations, for example, it used a retrospective approach and as such data on sunlight exposure, diet, physical activity and smoking status were not readily available. However, there was no seasonal variation in 25(OH)D levels, and it is known that 94% of patients with obesity eligible for WLS have insufficient 25(OH)D [34] irrespective of sunlight availability.

## 5. Conclusions

This study has identified that baseline 25(OH)D level is an important indicator for T2DM remission 12-months post-SG. It also highlights that increased levels of circulating 25(OH)D may prevent weight regain, despite it not being associated with weight loss post-surgery, and improve glucose homeostasis via the lipid profile. In addition, the importance of monitoring levels of 25(OH)D both pre- and post-WLS, particularly for patients with T2DM, has been detailed in order to determine those at risk of weight regain post-surgery. As such, this study supports recommendations of 25(OH)D supplementation pre- and post-surgery [56].

## Figures and Tables

**Figure 1 nutrients-14-02052-f001:**
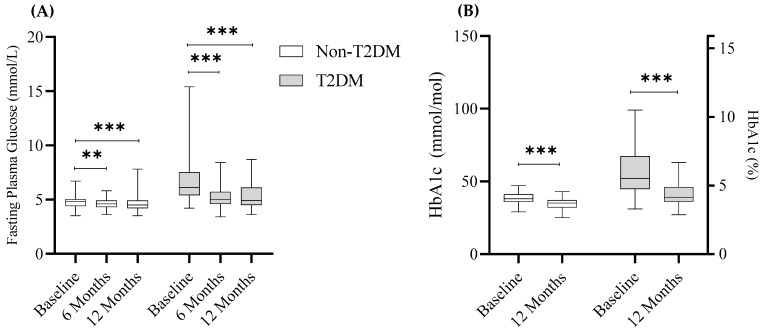
Effect of sleeve gastrectomy surgery on fasting plasma glucose and HbA1c levels. (**A**) Fasting plasma glucose levels (F. Glucose) at baseline, 6- and 12-months post-surgery for non-T2DM and T2DM groups. (**B**) Serum HbA1c levels at baseline and 12-months post-surgery for non-T2DM and T2DM groups. Data are presented as mean ± standard error of the mean. Statistical differences between time points were determined via a 2-tailed paired *t*-test: ** *p* < 0.01, *** *p* < 0.001.

**Figure 2 nutrients-14-02052-f002:**
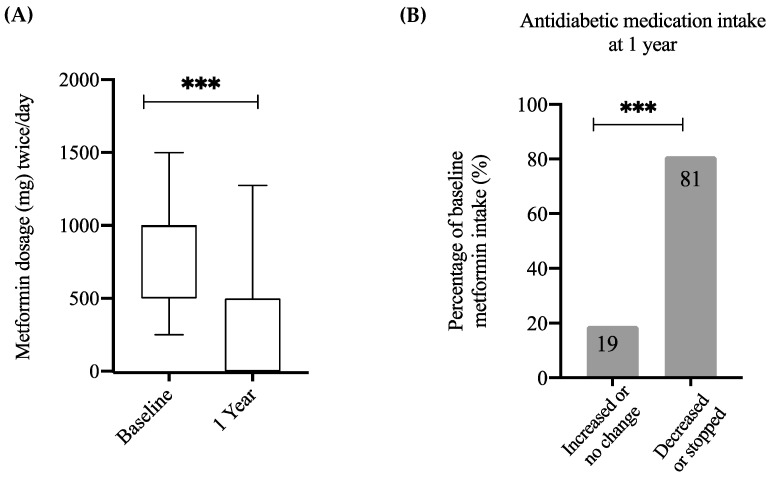
Changes in anti-diabetic medication intake post-surgery. (**A**) Boxplot displaying the distribution of Metformin intake at baseline and one-year post-surgery among patients with T2DM. (**B**) Percentage distribution of patients with T2DM at baseline who increased or did not change their dose of Metformin and those who stopped or reduced their dose of Metformin. Statistical differences between dose at baseline and dose at one year were obtained via 2-tailed paired *t*-test whereas statistical differences between subgroups of qualitative variables were obtained using chi-squared and McNemar tests: *** *p* < 0.001.

**Figure 3 nutrients-14-02052-f003:**
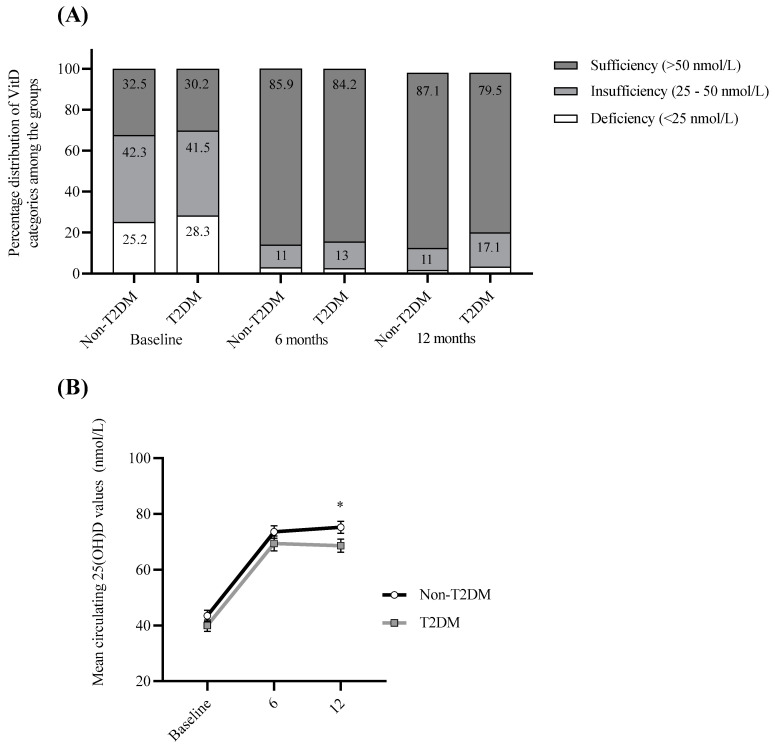
Summary of circulating 25(OH)D improvements. (**A**) Percentage distribution of T2DM and non-T2DM participants with deficient (<25 nmol/L), insufficient (25–50 nmol/L) and sufficient (>50 nmol/L) circulating 25(OH)D levels at baseline, 6-months and 12-months post-surgery for T2DM and non-T2DM groups. (**B**) Mean circulating 25(OH)D values at baseline, 6-months and 12-months post-surgery for T2DM and non-T2DM groups. Statistical differences between non-T2DM and T2DM cohorts were analyzed using 2-tailed independent samples *t*-test. * *p* < 0.05.

**Figure 4 nutrients-14-02052-f004:**
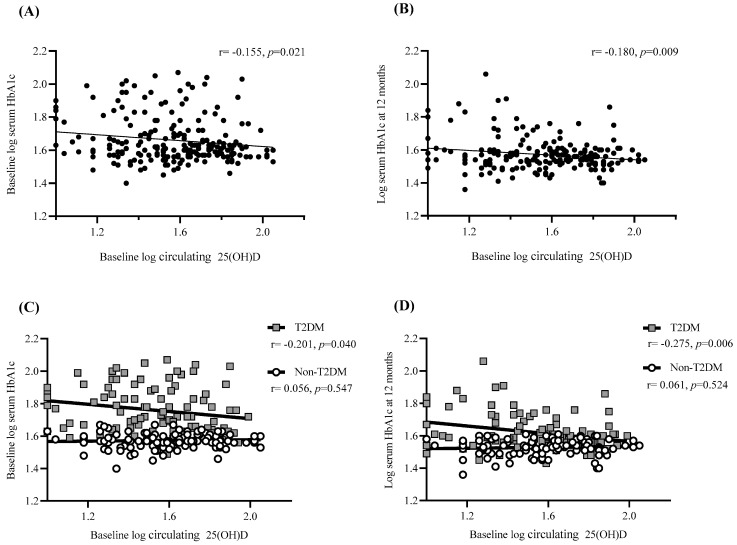
Association of 25(OH)D levels with weight regain and pre/post-surgical HbA1c. Scatter plot showing the correlation between (**A**) baseline 25(OH)D levels and baseline HbA1c for total cohort (*n* = 223), (**B**) baseline 25(OH)D levels and HbA1c levels at one-year post-surgery for total cohort (*n* = 209) (**C**) baseline 25(OH)D levels and baseline HbA1c depending on T2DM status (*n* = 105 for T2DM and *n* = 118 for non-T2DM)) and (**D**) baseline 25(OH)D levels and HbA1c levels at one-year post-surgery depending on T2DM status (*n* = 98 for T2DM and *n* = 111 for non-T2DM)). Linear trend line is shown with Pearson correlation statistic (r) and significance (*p*). *p* is considered significant if <0.05. Data were log-transformed prior to correlation analysis to improve normality.

**Figure 5 nutrients-14-02052-f005:**
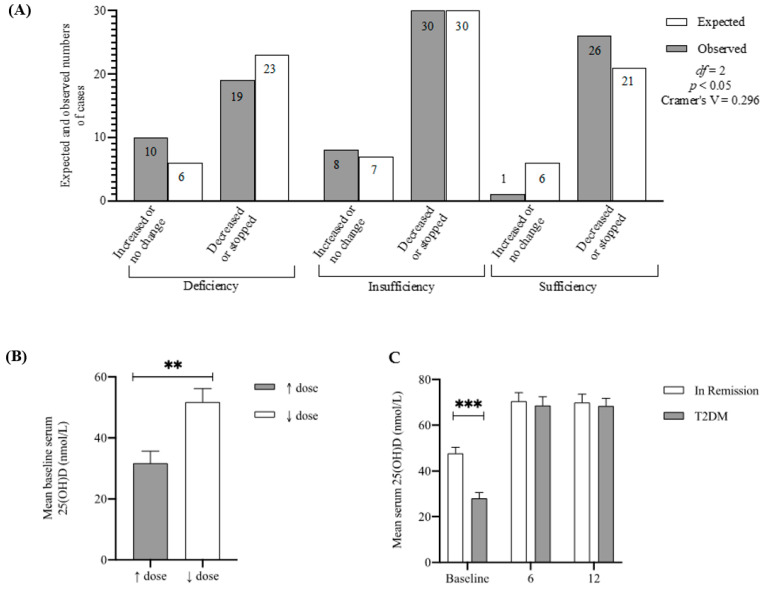
Baseline 25(OH)D Predicts T2DM Remission One-Year Post-Surgery. (**A**) Expected and observed numbers of participants who either increased/did not change their anti-diabetic medication dose or decreased/stopped their medication at 12 months post-surgery, according to their baseline 25(OH)D status. (**B**) Bar chart showing mean baseline circulating 25(OH)D values for patients who increased/did not change their anti-diabetic medication dose (↑ dose) (*n* = 20), or decreased/stopped their medication (↓ dose) (*n* = 74) at one-year post-surgery. Data are displayed as mean ± standard error of the mean. Statistical mean differences were determined by 2-tailed independent samples *t*-test (Mann-Whitney U). ** *p* < 0.01. (**C**) Bar chart showing mean circulating 25(OH)D values at baseline, 6 months and one-year post-surgery for patients who were in remission (*n* = 60) and patients who remained diabetic (*n* = 48). Data are displayed as mean ± standard error of the mean. Statistical differences between the groups were analyzed using 2-tailed independent samples *t*-test. *** *p* < 0.001.

**Table 1 nutrients-14-02052-t001:** Starting demographics of non-T2DM and T2DM cohorts.

	Non-T2DM *n* = 163		T2DM *n* = 146		Non-T2DM vs. T2DM
	Mean ± SD	Min–Max	Mean ± SD	Min–Max	*p* Value
Age (years)	46.32 ± 9.93	26–70	48.2 ± 10.6	23–70	0.261
Height (m)	1.66 ± 0.1	1.48–1.91	1.68 ± 0.1	1.45–1.91	0.076
Pre-Dietary Intervention Weight (kg)	144.59 ± 22.86	98–216	144.21 ± 22.9	94–225	0.884
Baseline Weight (kg)	129.58 ± 21.4	88–187	130.7 ± 21.27	84–208	0.647
BMI (kg/m^2^)	52.57 ± 6.65	36–74	51.15 ± 7.1	39–75	0.068
IBW (kg)	78.84 ± 7.73	57–91	71.67 ± 7.93	57–91	0.079
Excess Weight (kg)	75.75 ± 19.1	32–143	73.54 ± 20	36–134	0.545
% Excess Weight Loss (%)	19.45 ± 13.14	−20–63	18.58 ± 11.92	−6–58	0.545

Data are presented as means ± standard deviation (SD). Statistical differences between non-T2DM and T2DM patients were determined by 2-tailed independent *t*-tests; significance *p* values are shown in the right-hand column. Pre-dietary intervention weight refers to the heaviest weight at the first appointment prior to being given nutritional advice, all other measurements including baseline weight were taken on the day of surgery. BMI: body mass index; IBW: ideal body weight (calculated as if BMI was 25). % Excess weight loss is the percentage of baseline excess body weight that was lost prior to the surgery (calculated from the heaviest weight).

**Table 2 nutrients-14-02052-t002:** Comparison between biological characteristics of T2DM and non-T2DM patients at baseline, 6 and 12 months after SG.

	Baseline			6 Months			12 Months		
	Non-T2DM	T2DM		Non-T2DM	T2DM		Non-T2DM	T2DM	
	Mean ± SD	Mean ± SD	*p* Value	Mean ± SD	Mean ± SD	*p* Value	Mean ± SD	Mean ± SD	*p* Value
25(OH)D (nmol/L)	43.47 ± 22.77	39.96 ± 21	N.S	73.54 ± 25.3	69.43 ± 26.28	N.S	75.45 ± 25.47	68.66 ± 25.6	<0.05
FPG (mmol/L)	4.85 ± 0.67	7.91 ± 4.39	<0.005	4.72 ± 0.8	5.92 ± 2.41	<0.005	4.59 ± 0.7	5.84 ± 2.37	<0.005
HbA1c (mmol/mol) (%)	38.09 ± 4 (5.6 ± 2.5)	58.59 ± 18.87 (7.5 ± 3.9)	<0.005	NA	NA	N.S	34.28 ± 3.87 (5.3 ± 2.5)	43.59 ± 12.85 (6.1 ± 3.3)	<0.005
Cholesterol (mmol/L)	5.13 ± 0.96	4.35 ± 0.95	<0.005	4.89 ± 0.95	4.3 ± 0.95	<0.005	5.03 ± 0.95	4.6 ± 1	<0.005
HDL-Chol (mmol/L)	1.33 ± 0.32	1.15 ± 0.33	<0.005	1.44 ± 0.44	1.35 ± 0.45	N.S	1.66 ± 0.38	1.45 ± 0.4	<0.005
TG (mmol/L)	1.51 ± 0.71	1.86 ± 0.9	<0.005	1.34 ± 0.8	1.35 ± 0.54	N.S	1.16 ± 0.53	1.59 ± 0.76	<0.005
Chol:HDL	4.02 ± 1	4 ± 1.25	N.S	3.61 ± 1.1	3.39 ± 1	N.S	3.15 ± 0.87	3.37 ± 1.1	N.S
LDL-Chol (mmol/L)	3.11 ± 0.86	2.29 ± 0.97	<0.005	2.88 ± 0.85	2.3 ± 0.86	<0.005	2.86 ± 0.89	2.45 ± 0.95	<0.005
AST (U/L)	77.25 ± 21.91	80.59 ± 25.46		74.7 ± 30	68.27 ± 16.48	<0.05	70 ± 22.54	72.89 ± 21.23	N.S
ALT (U/L)	23.28 ± 13.66	28.51 ± 18.87	<0.005	15.91 ± 15.32	18.49 ± 16	<0.05	13.47 ± 5.57	15.14 ± 7	<0.05
CRP (mg/L)	12.1 ± 9.7	10.88 ± 12.8	N.S	NA	NA	N.S	NA	NA	N.S
* CRP + 1 (mg/L)	30.65 ± 32.69	26.91 ± 22.7	N.S	NA	NA	N.S	NA	NA	N.S
fT4 (ug/dL)	15.74 ± 3.98	16.24 ± 5.46	N.S	NA	NA	N.S	15.82 ± 4.2	16.22 ± 4.5	N.S
TSH (mIU/L)	2.37 ± 2	2.23 ± 1.7	N.S	NA	NA	N.S	2.2 ± 4.11	1.57 ± 1.1	N.S
Systolic BP (mmHg)	140.11 ± 17.2	142 ± 18.74	N.S	NA	NA	N.S	129.88 ± 19.1	131.8 ± 18.44	N.S
Diastolic BP (mmHg)	76.29 ± 10.47	76.94 ± 10.95	N.S	NA	NA	N.S	74.65 ± 10.41	75 ± 10.75	N.S

Data are presented as mean ± standard deviation of the mean. HDL: high-density lipoprotein, TG: triglycerides, Chol:HDL: cholesterol ratio to HDL, LDL: low-density lipoprotein, AST: aspartate transaminase, ALT: alanine transaminase. CRP: C-reactive protein, T4: thyroxine, TSH: thyroid-stimulating hormone, BP: blood pressure. * CRP + 1 is the level of CRP one day after the surgery. Mean difference between non-T2DM and T2DM was obtained using 2-tailed independent samples *t*-test.

**Table 3 nutrients-14-02052-t003:** Comparison of Pre-Surgery Factors in Participants with and without Diabetes Remission.

Factor	No Remission at 12 Months (*n* = 56)	Remission at 12 Months (*n* = 75)	*p* Value
Sex (F/M)	38/18	48/27	0.646
Age (years)	51.5 ± 8.4	49.4 ± 10.7	0.228
BMI (kg/m^2^)	50.47 ± 7.4	51.1 ± 6.3	0.615
EWL (%)	17.2 ± 13.2	20.7 ± 11	0.098
25(OH)D (nmol/L)	28.1 ± 16.1	47.5 ± 21.1	<0.001 ***
FPG (mmol/L)	10 ± 5.5	6.6 ± 3.1	<0.001 ***
HbA1c (mmol/mol) (%)	70.2 ± 20.8 (8.6 ± 4.1)	50.3 ± 13.5 (6.8 ± 3.4)	<0.001 ***
Cholesterol (mmol/L)	4.1 ± 1	4.4 ± 1	0.084
HDL-Chol (mmol/L)	1.1 ± 0.3	1.2 ± 0.3	0.055
TG (mmol/L)	2 ± 1	1.8 ± 0.9	0.092
Chol:HDL	3.9 ± 1.2	3.9 ± 1.2	0.893
LDL-Chol (mmol/L)	2.1 ± 0.9	2.3 ± 1	0.267
AST (U/L)	89.1 ± 30.1	74.3 ± 19.1	0.002 **
ALT (U/L)	27.3 ± 17.8	28.2 ± 19	0.527
T4 (ug/dL)	16.1 ± 5.1	16.6 ± 5.8	0.759
TSH (mIU/L)	2.41 ± 1.7	1.94 ± 1.3	0.749
Systolic BP (mmHg)	137.3 ± 20.7	144.2 ± 17.5	0.087
Diastolic BP (mmHg)	74.8 ± 12	78 ± 10.6	0.165
Metformin Dose (mg/daily) *	979.2 ± 286	813.64 ± 306.2	0.004 **
Hypertension (case no.)	46 (82.1)	49 (65.3)	0.033 *
Dyslipidaemia (case no.)	20 (35.7%)	23 (30.7%)	0.543

Pre-surgery factors in participants with and without T2DM remission were compared in order to identify factors that may impact remission at 12 months. EWL: excess weight loss, 25(OH)D: circulating vitamin D, FPG: fasting plasma glucose, HbA1c: Glycated hemoglobin, HDL-chol: high-density lipoprotein, TG: triglycerides, Chol:HDL: cholesterol ratio to HDL, LDL-chol: low-density lipoprotein, AST: aspartate transaminase, ALT: alanine transaminase, T4: thyroxine, TSH: thyroid-stimulating hormone, BP: blood pressure. * Metformin average daily dosage (250–2550 mg/daily) was noted in all participants pre-surgery and in remission. For continuous variables, either the two-tailed independent samples *T*-test or Wilcoxon signed-rank test was used to compare mean differences between two independent groups for parametric and non-parametric data, respectively. Comparisons between categorical groups and percentages were calculated by Chi-square (χ^2^) and McNemar tests. * *p* < 0.05, ** *p* < 0.01, *** *p* < 0.001.

**Table 4 nutrients-14-02052-t004:** Uni- and Multivariate Logistic Regression Analysis to Evaluate the Influence of Different Pre-Surgery Factors on T2DM Remission at 12 months.

Factor		β_0_	β_1_	R^2^	OR	95% CI	*p*-Value *
Univariate Analysis							
Age		−1.418	0.022	0.015	1.023		0.227
BMI							
HbA1c		−4.186	0.067	0.330	1.069	1.041–1.098	<0.001
25(OH)D		1.665	−0.056	0.278	0.954	0.920–0.971	<0.001
FPG		−2.043	0.225	0.193	1.252	1.107–1.417	<0.001
AST		−2.523	0.028	0.117	1.028	1.010–1.046	0.002
Metformin Dose		−2.038	0.002	0.095	1.002	1.001–1.003	0.006
Hypertension		−0.956	0.892	0.047	2.441	1.061–5.614	0.036
Multivariate Analysis							
Model I	25(OH)D	0.233	−0.056	0.379	0.946	0.918–0.974	<0.001
	FPG		0.173		1.189	1.048–1.349	0.007
Model II	25(OH)D	−3.287	−0.064	0.445	0.938	0.909–0.969	<0.001
	FPG		0.203		1.225	1.081–1.389	0.001
	Age		0.071		1.073	1.015–1.135	0.013

Univariate logistic regression was used to determine which pre-surgery factors may impact T2DM remission. Following this, multivariate logistic regression was used to determine which of these were able to predict T2DM remission 12-months post-surgery. Continuous variables were references to their median value. For categorical variables, the reference category was hypertension (yes). * *p*-value for the Wald statistic. β_0_ = constant of the model; β_1_ = coefficient of the explicative variable; R^2^ = Nagelkerke statistic; OR = odds ratio; CI = confidence interval; FPG = fasting plasma glucose.

## Data Availability

These data were based on hospital outpatient clinics, and are not available to share. Specific questions or requests can be directed to the corresponding author.

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
