# Peer review of "Vitamin D Levels as an Important Predictor for Type 2 Diabetes Mellitus and Weight Regain Post-Sleeve Gastrectomy"

_nutrients, 2022, doi:10.3390/nu14102052_

Round 1

Reviewer 1 Report

Authors presnet the work about the Vitamin D Levels as an predictor for T2DM and 2 Weight Regain Post-Sleeve Gastrectomy. Despite the significance of the issues presented, I have some doubts as to the accuracy of all the problems raised in the work.

Vitamin D is produced in the skin in response to ultraviolet (sunlight) exposure. Subsequently, it is hydroxylated at positions 25 and to gain full hormonal activity. On the other hand, 25-OH vitamin D [25(OH)D3] is a good marker of vitamin D status. The kidney, brain, bone, skin, prostate, and white blood cells can convert 25(OH)D3 to its active form [1,25(OH)2D3]. It can be anticipated that low serum levels of 25(OH)D3 will limit the synthesis of the active form in all these tissues. Serum 25(OH)D3 levels are mainly determined by the exposure to sunlight and vitamin D supplementation. In addition, higher fat tissue content is associated with lower serum 25(OH)D3 levels, possibly because of its ability to store vitamin D. Vitamin D metabolism is involved in the formation of other metabolites, such as 3-epi-25(OH)D3 and 24,25(OH)2D which are not considered to be physiologically active. However, based on recent studies, they play an important role in the regulation of general metabolism (3-epi-25(OH)D3 levels are associated with an improved cardiovascular risk profile, 3-epi-1,25(OH)2D3, derived from 3-epi-25(OH)D3, effectively reduces blood parathormone without inducing changes in the plasma calcium levels, 24,25(OH)2D3, consider to be an inactive form of vitamin D, protects the cell from 1,25(OH)2D3 toxicity and modulates the antioxidant potential by binding catalase, 24,25(OH)2D3 plays an important role in normal bone integrity, function, and healing. So why authors only present one of the vitamin D metabolites. I understend that this study investigated whether 25(OH)D levels could predict metabolic improvements in patients who underwent SG. But we should have in minde that there are much more interaction that have impact on it.

Introduction

Authors write that several studies have demonstrated the positive role for 25(OH)D in adipose tissue metabolism - it should be clearly presented. Additionally, nutritional mistakes and low exposure to sunlight should be shown as the main cause of low vitamin D concentration in presented population? On the other hand it has been checked?

Some data indicate that reduction on fatty tissue with out any other supelementation interactions may lead to increased Vit D serum concentraction.

Disscusion:

Authors write thaht observed correlations between baseline 25(OH)D and (1) HbA1c levels, (2) antidiabetic medication intakes and (3) lipid profile, may support and explain the finding that baseline 25(OH)D is able to predict the presence of T2DM one-year post-surgery. But observed situation is due the fact that it is a normal situation in this population

Certain limitations in presented work may unfortunately,  have a significant influence on the obtained results, which cannot be ruled out at this stage of study.

Author Response

The authors would like to thank the reviewers for their comments and considerations. We will respond to each point in turn below.

Point 1: I Why authors only present one of the vitamin D metabolites. I understand that this study investigated whether 25(OH)D levels could predict metabolic improvements in patients who underwent SG. But we should have in mind that there are much more interaction that have impact on it.

Response 1: We appreciate your consideration of the manuscript and acknowledge that insight into additional metabolites would be of interest, and would make three main points in response: firstly, for this clinical study the authors concentrated on 25(OH)D, as the metabolites referred to by the reviewer were considered to be more suited to in vitro analysis, due to the indirect impact and influence they may have, and this was considered beyond the scope of this current study. Secondly, whilst beyond the scope of this paper we are currently undertaking work to explore the in vitro effects of vitamin D metabolites on human adipocyte inflammation, which will add to the current in vitro studies on this topic to date (additional studies on this topic are referred to below this text). Finally, as this was a retrospective study, in which the authors collected the data from the registry of routinely measured parameters from the hospital, the authors were not able to measure additional analytes on this occasion. 

The authors would also detail that five different statistical analyses were undertaken to confirm their findings, and all tests gave similar results (Chi-Square (X2) or McNemar tests, one-way ANOVA or T-test or Wilcoxon Signed-Rank Test, Bivariate Pearson correlation analysis, simple, multiple and logistic regression analysis, ANN model development and sensitivity analysis)

Additional studies on Vitamin D metabolites in adipocytes:

  1. Mechanisms by Which Vitamin D Prevents Insulin Resistance and Associated Disorders: https://www.ncbi.nlm.nih.gov/pmc/articles/PMC7554927/
  2. Vitamin D reduces the inflammatory response and restores glucose uptake in adipocytes: https://onlinelibrary.wiley.com/doi/10.1002/mnfr.201200383
  3. Vitamin D and adipogenesis: https://academic.oup.com/nutritionreviews/article/66/1/40/1919558

Point 2: Authors write that several studies have demonstrated the positive role for 25(OH)D in adipose tissue metabolism - it should be clearly presented.

Response 2: We thank the reviewer for their considerations, and we have now amended our introduction in response to detail further studies on adipose tissue metabolism (tracked).

Point 3: Additionally, nutritional mistakes and low exposure to sunlight should be shown as the main cause of low vitamin D concentration in presented population? On the other hand it has been checked?

Response 3: Thank you for this comment, in this study, we were only able to collect the dates of the blood collections, rather than using a sunlight questionnaire from individuals. However, noting that typically people are more exposed to the sun in the summer, we did review the blood collection data, which showed no significant variation in vitamin D across the four seasonal groups observed. Data detailing these findings are in section 3.1 of the paper. The lack of substantial difference noted by our data may relate to level of sunshine in the UK, more awareness of sunscreen and less overall variation in exposure to the sun over different seasons.

Point 4: Some data indicate that reduction on fatty tissue without any other supplementation interactions may lead to increased Vit D serum concentration.

Response 4: We acknowledge this comment, however this study was examining the effect of vitamin D levels on metabolic improvements regardless of how vitamin D levels were increased.

Point 5: Authors write that observed correlations between baseline 25(OH)D and (1) HbA1c levels, (2) antidiabetic medication intakes and (3) lipid profile, may support and explain the finding that baseline 25(OH)D is able to predict the presence of T2DM one-year post-surgery. But observed situation is due the fact that it is a normal situation in this population.

Response 5: For all weight loss surgeries, some patients benefit more than other patients, and this paper was interested in what factors may influence or predict outcome benefits. Why some individuals failed to maintain their weight loss or achieve diabetes remission and how can we predict this.

Point 6: Certain limitations in presented work may unfortunately, have a significant influence on the obtained results, which cannot be ruled out at this stage of study.

Response 6: This is the challenge for observational and or retrospective clinical studies, as the   authors sought to use the data available from the clinic, with noted complex clinical histories in such tertiary care centres. The authors utilised several statistical methodologies and collected more than 80 variables to adjust for potential confounders.

Reviewer 2 Report

I would recommend to cite more current sources, namely ones that were published during the last two years.

Author Response

Response to Reviewer 2 Comments

The authors would like to thank the reviewers for their comments and considerations. 

 Point 1: I would recommend to cite more current sources, namely ones that were published during the last two years.

Response 1: Thank you for your comment, the references have now been updated in the manuscript (tracked)

Reviewer 3 Report

The article is well written, it is an assessment of the relationship between vitamin D and the prognosis of bariatric surgery. Congratulations for such an innovative article.

The subject is of great interest since one of the alternatives for the treatment of diabetes is bariatric surgery, since the origin of diabetes, although multifactorial, depends on obesity.

I find this approach to a factor (in this case, vitamin D) that can be easily measured and that offers a true and exact prognosis of post-gastrectomy evolution to be of great interest. These types of clinical articles are also of interest to specialized readers in this field.

Due to the length of this article, I find the separation between the data and the supplementary material appropriate, whose choice seems appropriate to me as well.

Author Response

Response to Reviewer 3 Comments

The authors would like to extend their appreciation for reviewer's comments and considerations.